# Pain as a Protective Factor for Alzheimer Disease in Patients with Cancer

**DOI:** 10.3390/cancers15010248

**Published:** 2022-12-30

**Authors:** Siqi Xia, Xiaobo Yu, Gao Chen

**Affiliations:** 1Department of Neurosurgery, Second Affiliated Hospital, School of Medicine, Zhejiang University, Hangzhou 310003, China; 2Key Laboratory of Precise Treatment and Clinical Translational Research of Neurological Diseases, Zhejiang University, Hangzhou 310003, China; 3Clinical Research Center for Neurological Diseases of Zhejiang Province, Hangzhou 310003, China

**Keywords:** cancer, Alzheimer disease, pain, risk factor, SEER, mortality, cancer site, survival year

## Abstract

**Simple Summary:**

Alzheimer disease (AD) and cancer have been reported to be inversely correlated in epidemiological studies. However, the mechanism behind it is not clear. The aim of our retrospective study was to assess the 11 risk factors, including pain, for subsequent AD death in patients with cancer. We examined a SEER Research Plus population of 25,512 cases and 127,560 controls. We found that pain was related to lower AD risk in all subgroups except for digestive cancer. In addition, age, sex, race, number of in situ/malignant tumors, number of benign/borderline tumors, cancer site, cancer-directed surgery, radiation, chemotherapy and survival years were independent factors of AD risk in cancer patients. The risk factors varied by cancer site and race. This study demonstrated pain as a novel protective factor of AD and suggests the uniqueness of the digestive system in interacting with the central nervous system, which provide new perspectives for future studies.

**Abstract:**

Objective: Alzheimer disease (AD) and cancer have been reported to be inversely correlated in incidence, but the mechanism remains elusive. Methods: A case-control study was conducted, based on the SEER (Surveillance, Epidemiology, and End Results) Research Plus data, to evaluate 12 factors in patients with cancer. Results: Severe pain was related to reduced AD risk, while older age at cancer diagnosis, female, longer survival years after tumor diagnosis, more benign/borderline tumors, less cancer-directed surgery, and more chemotherapy were associated with higher AD risk. In addition, patients of different races or with different cancer sites were associated with different risks of getting AD. Cases had a higher prevalence of severe pain than controls in all race and cancer site subgroups, except for in digestive cancer, where the result was the opposite. Conclusions: This study indicated pain as a novel protective factor for AD in patients with cancer. The mechanism behind it may provide new perspective on AD pathogenesis and AD-cancer association, which we discussed in our own hypothesis of the mechanism of pain action. In addition, digestive cancer pain had an opposite impact on AD risk from other cancer pains, which suggests the uniqueness of digestive system in interacting with the central nervous system.

## 1. Introduction

Alzheimer disease (AD) and cancer are among the leading causes of human death around the world [1]. Both are age-related diseases possibly with some common molecular pathways in their pathogenesis [2]. However, there is still a lack of satisfactory understanding of their mechanisms, which to some extent impedes the development of effective treatments for each disease [3]. Several prior epidemiological studies have demonstrated a relationship between AD and cancer: cancer survivors had a lower risk of AD than cancer-free people [4,5,6,7,8], cancer risk was lower among AD patients than those without AD [9,10,11,12], and survivors of certain types of cancer, such as prostate cancer, were found to have a slightly elevated AD risk [13]. However, the biological mechanism behind the association remains widely debated. Risk factors that have been studied previously include age, race, sex, comorbidities [14], and cancer types. Existing main biological theories include the Warburg effect theory, the two-hit hypothesis theory, the unfolded protein response theory, chronic inflammation, age-related metabolic deregulation, epigenetic causes, family history, and exogenous infection [2,15,16]. However, there have been no studies on pain and AD risk in cancer patients. Four previous studies reported the positive association between non-cancer pain and subsequent AD, but most of them attributed it to the mediation of mood or sleep disorders caused by pain [17,18,19,20]. A new perspective on pain and other risk factors in the etiopathology of cancer and AD may provide new chances for effective therapeutics.

The objective of this study was to examine pain and other risk factors for AD in cancer patients, both in the total sample and in subgroups. In this article, we calculated the prevalence, single factor difference, and odds ratio (OR) in an univariable and multivariable logistic regression model for established and novel risk factors, including pain. Then we analyzed them stratified by race and cancer site, which delineated the unique risk patterns for different race and cancer site groups, especially for digestive cancer. We also examined the predictive accuracy, sensitivity, and specificity of the logistic regression models by constructing receiver operating characteristic (ROC) curves.

## 2. Materials and Methods

In this case-control study, we used the Surveillance, Epidemiology, and End Results (SEER) Research Plus Data in 8 registries from 1975 to 2019 to obtain study samples. The SEER program, which is supported by the National Cancer Institute (NCI), collects cancer data from various state registries that cover approximately 35% of the US population. Since the SEER data is deidentified, this study is exempt from full board review by the institutional review boards at the participating research institutions. The study followed the Strengthening the Reporting of Observational Studies in Epidemiology (STROBE) reporting guideline.

### 2.1. Study Populations

We included patients diagnosed at the age of over 20 with all primary cancers other than neural, head, face, or neck cancer between 1975 and 2019. We defined cancers using the International Classification of Diseases for Oncology, Version 3 (ICD-O-3)/WHO 2008. Only cases that were confirmed microscopically at diagnosis and by autopsy or death certificate at death were selected. We exclude those having brain or perineural invasion. The cause of death (COD) in the case group was Alzheimer disease defined by ICD-9 and 10. In the control group, the COD was defined besides its own or other cancer/neoplasm, AD, cerebrovascular disease (CVD), and all kinds of social events (e.g., accidents, suicide, homicide). We initially identified 29,040 cases and 440,355 controls, including 62 types of subdivided cancers. Then, we excluded those with invalid race, cancer-direct surgery, radiation data, those with age at death ≤65, and those with malignant behavior code but survival year <1. We selected control samples through random digit dialing to make case and control group 1:5 matched. The final sample had 25,512 cases and 127,560 controls. Figure 1 shows the flowchart of this study.

### 2.2. Outcome, Exposure, and Risk Factors

We chose the candidate variables based on previous literature and our biological inference. The corresponding data for case and control groups were then collected from SEER’ s electronic system. After calculation and categorization, we identified 11 potential risk factors, including age, sex, race (White, Asian/Pacific Islander, Black, American Indian/Alaska Native), pain rating (I, II), total number of in situ/malignant tumors (number 1), total number of benign/borderline tumors (number 2), cancer site, radiation therapy (Yes, No), cancer-directed surgery (Yes, No), chemotherapy (Yes, No/Unknown), and survival years after primary tumor diagnosis. The pain rating I was defined as in situ behavior, while pain rating II was defined as malignant behavior with survival years >1. The initial 62 cancer sites were further divided into 11 types according to the organ system, including bone and joint, breast, digestive, endocrine, hematological, Kaposi sarcoma, mesothelioma, miscellaneous, respiratory, skin and soft tissue, and urogenital cancers. The main outcome is whether or not death is due to AD.

### 2.3. Statistical Analysis

Descriptive statistics were used to describe the baseline characteristics of our study samples, both in total and after stratification by pain rating, race, and cancer site. Unadjusted case-control association analyses of the 11 potential risk factors were assessed using independent t-tests for continuous variables (age, number 1, number 2, and survival years) and a Χ^2^ test for categorical variables (sex, race, pain rating, cancer site, radiation, cancer-directed surgery, and chemotherapy). Risk factors that were *p* < 0.05 were selected for subsequent regression analysis. Multivariable statistical analyses of 2-level risk factors, including sex, pain rating, radiation therapy, cancer-directed surgery, and chemotherapy were performed using multivariable logistic regression. Multivariable statistical analyses for multi-level risk factors race and cancer site were assessed also using multivariable logistic regression, but with dummy variables, in which White race and urogenital cancer were set as the reference categories. In these analyses, univariable logistic regression models for risk of AD were first constructed. Then the multivariable logistic models were constructed, with forward stepwise selection to retain only those factors that were *p* < 0.05. Unadjusted odds ratio (OR) and adjusted OR were calculated to account for the risk level.

Then, the abovementioned analyses were repeated to explore case-control associations stratified by race and cancer site. Only subgroups with a population greater than 800 in both case and control groups were included. They were: White, Asian/Pacific Islander, Black, breast cancer, digestive cancer, skin and soft tissue cancer, and urogenital cancer groups. Missing data were not calculated. Statistical analyses were performed using SPSS version 26.0 (SPSS, Inc., Chicago, IL, USA) software. All statistical tests were two-sided, and statistical significance was set at *p* < 0.05.

## 3. Results

### 3.1. Overall Sample Characteristics

There were 25,512 case patients and 127,560 control patients. The average age of the case group was older than that of the control group (mean (SD) age, 71.92 (9.76) vs. 71.72 (9.80) years; *p* < 0.001) (Table 1). The case group had a significantly larger proportion of female, White, and American Indian/Alaska Native compared with the control group (16,670 (65.3%) vs. 65,911 (51.7%); 23,355 (91.5%) vs. 106,957 (83.8%), 49 (0.2%) vs. 126 (0.1%)) (age, sex: *p* < 0.001) and a smaller proportion of male, Asian/Pacific Islander, and Black (Appendix A). A substantially lower rate of case individuals had pain rating II compared to the controls (23,394 (91.7%) vs. 119,391 (93.6%), *p* < 0.001). Case patients had a significantly smaller number of in situ/malignant tumors (1.17 (0.45) vs. 1.20 (0.50), *p* < 0.001), a significantly larger number of benign/borderline tumors (0.40 × 10^−2^ (0.07) vs. 0.18 × 10^−2^ (0.04), *p* < 0.001), and substantially longer survival years after first cancer diagnosis (15.62 (8.86) vs. 11.71 (8.16) years, *p* < 0.001). Cancer sites were significantly different between case and control groups (*p* < 0.001; Appendix A). Case group had a significantly higher rate of cancer-directed surgery or chemotherapy compared to control group. No difference was detected in radiation therapy between the 2 groups.

### 3.2. Pain and AD Risk

In multivariable analysis, pain rating II was found to be associated with a lower AD risk in cancer patients (aOR = 0.849; 95% CI, 0.805–0.896; *p* < 0.001) (Table 2). The adjusted ORs for age and female sex were 1.055 (95% CI, 1.053–1.057; *p* < 0.001) and 1.315 (95% CI, 1.269–1.362; *p* < 0.001). Compared to White, the adjusted Ors for Asian/Pacific Islander, Black, and American Indian/Alaska Native are 0.408 (95% CI, 0.384–0.434; *p* < 0.001), 0.670 (95% CI, 0.623–0.721; *p* < 0.001), and 2.179 (95% CI, 1.545–3.075; *p* < 0.001). AD risk in cancer patients was correlated with fewer in situ/malignant tumors (aOR = 0.764, 95% CI, 0.740–0.788; *p* < 0.001) and more benign/borderline tumors (aOR = 1.987, 95% CI, 1.573–2.509; *p* < 0.001). Compared with the reference urogenital cancer, AD risks in skin and soft tissue (aOR = 1.190; 95% CI, 1.114–1.272; *p* < 0.001) and breast cancer (aOR = 1.090; 95% CI, 1.047–1.136; *p* < 0.001) were higher, whereas AD risks in digestive (aOR = 0.784; 95% CI, 0.753–0.816; *p* < 0.001), hematological (aOR = 0.683; 95% CI, 0.597–0.781; *p* < 0.001), respiratory cancer (aOR = 0.533; 95% CI, 0.486–0.586; *p* < 0.001) were lower. The difference in AD risks between urogenital and bone and joint (aOR = 0.829; 95% CI, 0.416–1.652; *p* = 0.59), endocrine (aOR = 1.040; 95% CI, 0.897–1.204; *p* = 0.61), mesothelioma (aOR = 0.769; 95% CI, 0.170–3.474; *p* = 0.73), or miscellaneous cancer (aOR = 2.133; 95% CI, 0.484–9.402; *p* < 0.32) were not significant. The adjusted OR for cancer-directed surgery, chemotherapy, and survival years were 0.902 (95% CI, 0.857–0.951; *p* < 0.001), 1.450 (95% CI, 1.370–1.534; *p* < 0.001), and 1.096 (95% CI, 1.093–1.098; *p* < 0.001). Radiation therapy had no significant relationship with AD risk in total cancer patients. Figure 2A shows the area under the receiver operating characteristic curve (AUC) for AD risk in overall cancer patients was 0.700 (95% CI, 0.697–0.703). The Youden’s index’s best cutoff value was 0.295. The cut-off value with the highest Youden Index was 0.295, with sensitivity of 71.2% and specificity of 58.4%.

### 3.3. Patterns of Pain Rating

We further explored the different characteristics between the pain rating I and rating II groups (Table 3). In the case group, there were 2118 (8.3%) pain rating I patients and 23,394 (91.7%) pain rating II patients, while in the control group, there were 8169 (6.4%) pain rating I patients and 119,391 (93.6%) pain rating II patients. In both the case and control groups, there was a significant difference in the distribution of age, sex, number of in situ/malignant tumors, cancer site, radiation therapy, cancer-directed surgery, chemosurgery, and survival years between pain rating I and II patients. In both groups, pain rating II patients tended to be older, male, fewer in situ/malignant tumors, received more radiation therapy, less cancer-directed surgery, more chemotherapy, and shorter survival years compared to pain rating I patients.

### 3.4. Racial and Tumor Site Variation in AD Risk

Subgroup analyses stratified by race demonstrated the significant association of pain rating II (aOR = 0.862; 95% CI, 0.816–0.911), more in situ/malignant tumors (aOR = 0.751; 95% CI, 0.727–0.775), and cancer-directed surgery (aOR = 0.901; 95% CI, 0.853–0.952) with reduced AD risk in White cancer patients. On the contrary, female, more benign/borderline tumors, chemotherapy, and longer survival years after first cancer were associated with a higher risk of AD in White cancer patients. Compared with the reference urogenital cancer, skin and soft tissue (aOR = 1.205; 95% CI, 1.127–1.289) and breast cancer (aOR = 1.110; 95% CI, 1.064–1.157) were associated with an increased risk of AD in White, whereas in digestive (aOR = 0.848; 95% CI, 0.813–0.884; *p* < 0.001), hematological (aOR = 0.675; 95% CI, 0.588–0.775; *p* < 0.001) and respiratory cancers (aOR = 0.438; 95% CI, 0.397–0.483; *p* < 0.001) the association was inverse. Figure 2B shows the AUC for AD risk in White cancer patients is 0.658 (95% CI, 0.655–0.662), with the best cut-off value of 0.232 (sensitivity: 69.1% and specificity: 54.2%). In Asian/Pacific Islander cancer patients, AD risk was only statistically related to age (aOR = 1.092; 95% CI, 1.083–1.102), female gender (aOR = 1.701; 95% CI, 1.499–1.929), and survival years (aOR = 1.118; 95% CI, 1.107–1.129). The case group had a lower prevalence of pain rating II than control group, but the difference was not statistically significant. The AUC for AD risk in Asian/Pacific Islander cancer patients is 0.730 (95% CI, 0.717–0.744) (Figure 2C). In Black cancer patients, more in situ/malignant tumors (aOR = 0.641; 95% CI, 0.530–0.775) were associated with reduced AD risk, whereas age (aOR = 1.076; 95% CI, 1.066–1.086) and survival years (aOR = 1.121; 95% CI, 1.108–1.133) were associated with increased risk of AD. The case group had a lower proportion of pain rating II patients than the control group, but it was only significant in independent t test. The AUC for AD risk in Black cancer patients is 0.737 (95% CI, 0.721–0.753) (Figure 2D). Appendix A shows the distribution of risk factors in patients stratified by race. Appendix A shows the results of univariable and multivariable logistic regression.

Subgroup analyses stratified by cancer site demonstrated pain rating II as a protective factor for AD risk in breast (aOR = 0.772; 95% CI, 0.718–0.831) and skin and soft tissue cancers (aOR = 0.815; 95% CI, 0.710–0.934), but as a risk factor in digestive cancer (aOR = 1.130; 95% CI, 1.006–1.270) (Appendix A). The case group with urogenital cancer had a significantly lower proportion of pain rating II patients (9358 [98.1%] vs. 49,995 [98.5%], *p* = 0.006), but it was not included in the regression model (Appendix A). In all four types of cancers, Asian/Pacific Islander and Black had a lower risk of AD compared with the reference White group. The adjusted OR for Asian/Pacific Islander and black compared with White, respectively, were 0.394 (95% CI, 0.352–0.440) and 0.604 (95% CI, 0.531–0.686) in breast cancer, 0.461 (95% CI, 0.410–0.518) and 0.660 (95% CI, 0.568–0.767) in digestive cancer, 0.524 (95% CI, 0.331–0.829) and 0.359 (95% CI, 0.178–0.724) in skin and soft tissue cancer, and 0.382 (95% CI, 0.345–0.424) and 0.541 (95% CI, 0.483–0.607) in urogenital cancer. AD risk in all 4 cancers were also related to female, less in situ/malignant tumors, and longer survival years. The AUC for AD risk in the 4 cancers (breast, digestive, skin and soft tissue, urogenital) are 0.639 (95% CI, 0.632–0.645), 0.679 (95% CI, 0.671–0.687), 0.664 (95% CI, 0.649–0.679), and 0.667 (95% CI, 0.661–0.673) (Figure 2E–H).

## 4. Discussion

### 4.1. Pain as a Protective Factor for AD in Patients with Cancer

Despite progress in the study of AD, effective prevention or treatment for it is still lacking due to a lack of explicit understanding of its pathogenesis. In the last decade, there has been emerging research studying the AD-cancer association, aiming to find out the common and different mechanisms in these two intractable diseases [21]. Many epidemiological studies demonstrated the inverse relationship between AD and certain cancers, including a lower chance of developing cancer in AD patients and a lower chance of AD in cancer patients, whereas several other studies reported little or a positive association with certain cancers [22,23]. Main factors studied for AD risk in cancer patients include age, sex, cancer type, tumor stage, size, and grade, treatment, and comorbidities such as CVD [24]. In addition, there have been four studies reported about the positive relationship between AD and non-cancer chronic pain conditions (NCPCs) [17,18,19,20], and most of them attribute the cause to the patient’s depression/anxiety mood or sleep disorders. However, there is no literature regarding the association of pain with subsequent AD risk in cancer patients. 

In this study, we demonstrated that pain served as an independent protective factor for AD death. A meta-analysis revealed a high prevalence of pain in cancer patients, with 66.4% in advanced, metastatic, or terminal disease, 50.7% in all cancer stages, 39.3% after curative treatment, and 55.0% during anticancer treatment [25]. In addition, a meta-analysis found that progress in the pathophysiological mechanisms of pain and the wider use of antinociceptive therapies had not influenced the prevalence of pain in cancer patients [26]. In this study, we categorize pain into two ratings based on cancer behavior and survival years, representing both the degree and the duration of pain. The reason we take these two measurements of pain into account is that AD is a chronic disease, and the effect of any influencing factor in its pathogenesis, such as cancer pain, must have a certain degree and a long-lasting effect. Therefore, we categorize the cancer pain into rating I and II, which are defined as in situ cancer and malignant cancer with survival years >1 respectively. In both the total samples and all subgroups except for digestive cancer, cancer patients who died by AD had a lower prevalence of pain rating II compared to those who died by control causes. Among them, the group that had the most obvious relationship with cancer pain was breast cancer, with an OR of 0.772. A previous bioinformatic study recognized many shared transcription factors (TFs) and regulatory processes that were closely related to the adaptive immune response from dramatically different directions, which may play crucial roles in both AD and breast cancer pathogenesis [27]. We also found that digestive cancer patients had elevated AD risk by cancer pain, with an OR of 1.13. 

Overall, our result regarding pain was opposite to the four previous studies that reported a positive relationship between pain and AD risk. The reasons might be as follows. First, in the four previous studies, the discrepancy in mood or sleep disorder degree between case and control groups may have been larger than in this study since they used individuals with no NCPCs as controls. Second, the sources of NCPCs in those studies included headache, neuropathic pain, and inflammatory pain (such as osteoarthritis), which may have confounders on AD risk apart from the pain itself, such as the effects of the primary disease on the central nervous system and on peripheral inflammation [28]. Although we didn’t subdivide pain character due to the limitations of the SEER database in our study, we excluded cancers appearing on the nervous system, head, face, or neck, as well as those with perineural invasion or metastasis to the brain. Therefore, the association of pain with AD risk in our study may be less confounded by other influencing factors on the central nervous system.

### 4.2. Hypothesis on Cancer-AD Association

It has been debated whether the association between AD and cancer is the consequence of their shared biological mechanisms or the pharmacological treatments given to patients [2]. Recently a meta-analysis proved that the inverse association between AD and cancer may be possibly attributable to shared inverse etiological mechanisms or survival bias, but was not likely due to competing risks bias, diagnostic bias, or inadequate control for potential confounding factors [29]. Since we set age, survival years, and cancer treatments including radiation, therapy, and cancer-directed surgery as independent risk factors in our study, we thus make the hypothesis that pain itself may have a direct influence on AD pathogenesis and that it intertwines with the common inverse etiological mechanisms that AD and cancer share. Nowadays, there are diverse proposed biological theories trying to explain the inverse correlation between AD and cancer, such as the Warburg effect theory, the two-hit hypothesis theory, the unfolded protein response theory, chronic inflammation, metabolic deregulation hypothesis, and epigenetic causes [16], but none of them can completely explain the two diseases [2]. Therefore, there are also merging hypotheses which include a subset of molecules such as p53, Wnt, UPS, and PIN1 [2]. Based on our results and biological reference, we hypothesize that enduring pain in malignant cancer patients is related to inhibition of the systemic sympathetic nervous system, which can regulate whole-body metabolic homeostasis [30], central nervous system (CNS) neural activity, and immune cell activity [31], and thus have an impact on AD and cancer pathogenesis in two relatively opposite directions. This can explain why cancer patients who died by AD in our study had a lower proportion of individuals with the pain rating II. However, the specific molecular mechanisms involved need further experimental verification. Appendix A describes our hypothesis of the mechanism of pain action in AD and cancer.

Our hypothesis can also explain why digestive cancer doesn’t fall within the range of pain’s protective influence. The digestive system has long been found to be tightly connected to the central nervous system in many ways, such as the gut-brain axis, gut microbiota, and intestinal sympathetic nervous system. Therefore, pain may have little influence on digestive cancer-AD association, due to the strong direct link between digestive cancer and the CNS. The elevated risk caused by pain in digestive cancer patients might be explained by the effect of the local intestinal sympathetic nervous system and gut microbiota, which can affect the brain microenvironment directly through the gut-brain axis, increase neuroinflammation in the brain, and thus increase the risk of getting AD. In both the total samples and the subgroups that had a difference in distribution among the two pain ratings, the difference was significant, except in Asian/Pacific Islander, where the difference existed but was not significant (case vs. control: 92.9% vs. 93.4%, *p* = 0.51). We can also explain it by our hypothesis for that the average level of systemic sympathetic activity might be relatively low in Asian/Pacific Islander, which needs further verification [32]. This makes cancer pain less influential in AD prevention in Asian/Pacific Islander compared with other races, though the pain association remained.

### 4.3. Other Risk Factors for AD in Patients with Cancer

Apart from pain, we also found that age, female sex, American Indian/Alaska Native ethnicity, more benign/borderline tumors, chemotherapy, and longer survival years were associated with a higher risk of AD in cancer patients, whereas more in situ/malignant tumors and cancer-directed surgery were linked to a reduced AD risk. In fact, the latter two factors are, to some extent, similar to pain because both of them can increase a patient’s physical pain. Age, sex, and race have long been discussed as risk factors for AD [33]. Older age, female sex, and Black were usually recognized as risk factors for AD [34]. There has been debate on the effect of chemotherapy on AD incidence. A study found chemotherapy was independently related to decreased AD risk [13]. Another study reported that chemotherapy used in colorectal cancer survivals was associated with reduced risk for AD and other neurocognitive disorders [35]. However, chemotherapy-induced cognitive impairment was also been reported in studies [36].

### 4.4. Study Strengths

This study has strengths. To our knowledge, this is the first comprehensive investigational study exploring the influence of pain in the cancer-AD association. In addition, we proposed a new hypothesis about pain action and provided a new perspective on AD pathogenesis and the cancer-AD relationship. Another strength is the large size and proper setting of the study. More than 25,000 case and 120,000 control samples across the United States, with four main race categories, and 62 cancer types were included, which ensures the reliability of the study. In addition, we established strict inclusion and exclusion criteria to avoid potential confounding factors for the AD outcome, which ensures the study’ s rigorousness. We covered 11 widely-distributed risk factors including demographic characteristics, cancer features, and treatment information in our study, which allow us to explore cancer-AD relationship broadly. What’s more, we did subgroup analyses stratified by race and cancer site, which added depth and completeness to the study results. For the first time, we demonstrated the uniqueness of digestive cancer pain in AD pathogenesis compared with other cancers, which inferred the intricate interaction between the gastrointestinal system and the central nervous system through known or unknown mechanisms.

### 4.5. Study Limitations

This study also has limitations. First, it was based on a single database. Future multi-database and multicenter randomized controlled studies may be needed to verify the association between pain and AD. Second, the measurement of pain is semi-quantitative and indirect, which may lead to a loss of detailed evidence and residual confounding [29,37]. However, we think our grading of pain was enough to reflect the effect of pain degree and pain time on subsequent AD risk. Completely quantitative measurement of pain degree by detailed grading such as the 1–10 score system may be unnecessary. Additionally, the causes of pain in cancer patients may be extensive, such as physical pressure, neuropathic, treatment-induced, and comorbidity-induced pain. However, late-stage malignant cancer itself is the biggest source of pain, according to the literature [25,26]. Third, due to the limitation of variable items in the SEER database, other underlying or contributory causes of AD such as comorbidities [14], socioeconomic conditions, neuroimaging changes [38], genetic differences, histological nature of the cancer, and treatment for AD or pain were not included in the study, which may be potential confounders. Although they were not considered by us in our study and can be compromised by the large samples and exhaustive subgroup analyses, future studies based on a database that includes these variables may promote the research. In addition, the diagnosis sequence of cancer and AD is not known due to the lack of information at the baseline of our study. Although we did get a positive outcome showing an inverse association based on the limited data, the reverse association over time between AD and cancer can be explored in future studies.

We view our findings with caution. Although an association between cancer pain and subsequent AD risk was proved, causation cannot be inferred due to the nature of the case-control study. Further experimental study may be needed to address the exact causal relationship between pain and AD in cancer patients, as well as the molecular mechanisms behind it. Furthermore, we suggest that pain should be considered as a factor in studies regarding cancer-AD relationship in the future.

## 5. Conclusions

This study of pain and other risk factors for AD in cancer patients identified pain as a novel protective factor for AD in the total sample and all subgroups, except in digestive cancer, where it was a risk factor. We proposed a new hypothesis of pain action in AD and cancer pathogenesis. Our study provides a new perspective on the pathogenesis of AD and the mechanism behind the AD-cancer relationship. It also suggests the importance of recognizing pain as an influencing factor in future epidemiological studies of the cancer-AD association.

## Figures and Tables

**Figure 1 cancers-15-00248-f001:**
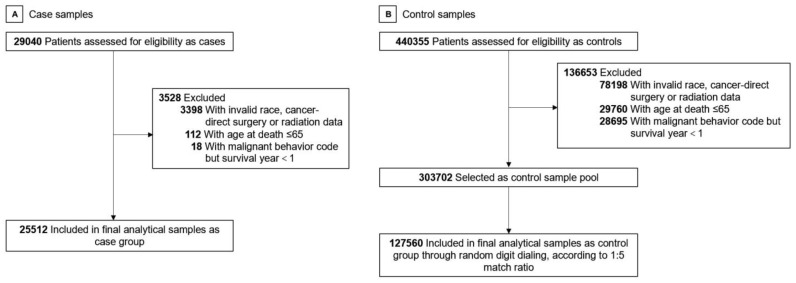
Study Flowchart. (**A**) Case sample selection. (**B**) Control sample selection.

**Figure 2 cancers-15-00248-f002:**
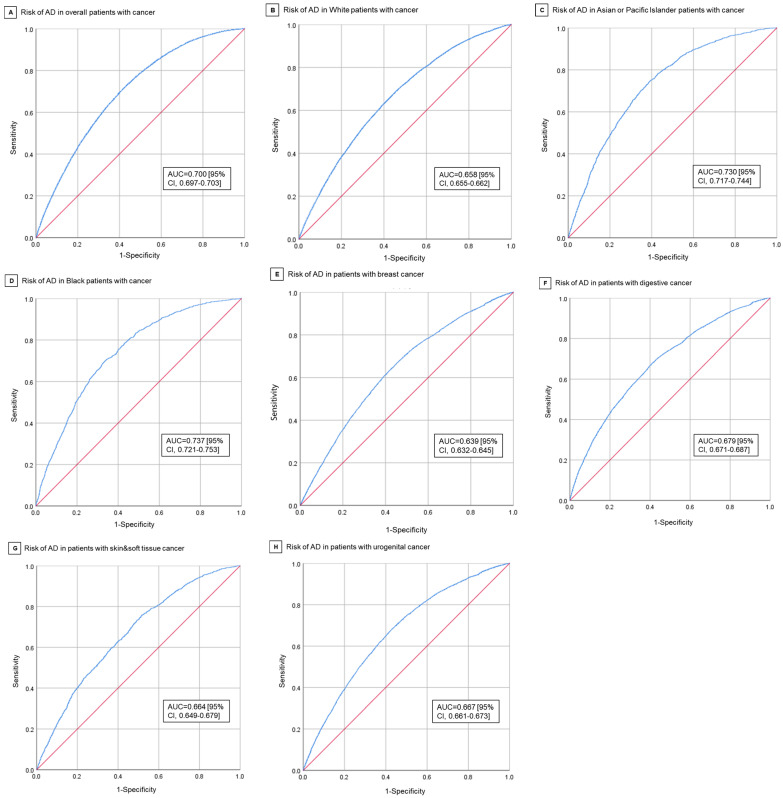
Receive-Operating Characteristics Curves for Risk of AD in Cancer Patients. (**A**) Risk of AD in overall cancer patients. (**B**) Risk of AD in White cancer patients. (**C**) Risk of AD in Asian/Pacific Islander cancer patients. (**D**) Risk of AD in Black cancer patients. (**E**) Risk of AD in breast cancer patients. (**F**) Risk of AD in digestive system cancer patients. (**G**) Risk of AD in skin and soft tissue cancer patients. (**H**) Risk of AD in urogenital cancer patients.

**Table 1 cancers-15-00248-t001:** Baseline characteristics of cases and controls.

Variable	Case Group, No. (%) (*n* = 25,512)	Control Group, No. (%) (*n* = 127,560)	*p*-Value
Age at cancer diagnosis, mean (SD), y	71.92 (9.76)	71.72 (9.80)	0.003 ^a^
Sex			<0.001 ^b^
Male	8842 (34.7)	61,649 (48.3)	
Female	16,670 (65.3)	65,911 (51.7)	
Race			<0.001 ^b^
White	23,355 (91.5)	106,957 (83.8)	
Asian/Pacific Islander	1217 (4.8)	13,254 (10.4)	
Black	891 (3.5)	7223 (5.7)	
American Indian/Alaska Native	49 (0.2)	126 (0.1)	
Pain rating ^c^			<0.001 ^b^
I	2118 (8.3)	8169 (6.4)	
II	23,394 (91.7)	119,391 (93.6)	
Total number of in situ/malignant tumors	1.17 (0.45)	1.20 (0.50)	<0.001 ^a^
Total number of benign/borderline tumors	0.40 × 10^−2^ (0.07)	0.18 × 10^−2^ (0.04)	<0.001 ^a^
Cancer site			<0.001 ^b^
Bone and Joint	10 (0)	65 (0.1)	
Breast	8390 (32.9)	30,030 (23.5)	
Digestive	5037 (19.7)	30,987 (24.3)	
Endocrine	243 (1)	1154 (0.9)	
Hematological	279 (1.1)	1924 (1.5)	
Kaposi Sarcoma	17 (0.1)	181 (0.1)	
Mesothelioma	2 (0)	15 (0)	
Miscellaneous	3 (0)	7 (0)	
Respiratory	525 (2.1)	6870 (5.4)	
Skin and Soft Tissue	1428 (5.6)	5557 (4.4)	
Urogenital	9578 (37.5)	50,770 (39.8)	
Radiation therapy			0.61 ^b^
Yes	6507 (25.5)	32,341 (25.4)	
No	19,005 (74.5)	95,219 (74.6)	
Cancer-directed surgery			<0.001 ^b^
Yes	23,269 (91.2)	112,996 (88.6)	
No	2243 (8.8)	14,564 (11.4)	
Chemotherapy			0.003 ^b^
Yes	1854 (7.3)	8613 (6.8)	
No/Unknown	23,658 (92.7)	11,8947 (93.2)	
Survival years after the first tumor diagnosis	15.62 (8.86)	11.71 (8,16)	<0.001 ^a^

Abbreviations: SD, standard deviation. ^a^
*p*-value by independent t-test. ^b^
*p*-value by Χ^2^ test. ^c^ Pain rating I indicates in situ behavior code; pain rating II indicates malignant behavior code, with survival years > 1.

**Table 2 cancers-15-00248-t002:** Analyses of risk factors for AD in cancer patients.

Variable	OR (95% CI) ^a^	*p*-Value	Adjusted OR ^b^ (95%CI)	*p*-Value
Age at cancer diagnosis, mean, y	1.055 (1.053–1.057)	<0.001	1.055 (1.053–1.057)	<0.001
Sex				
Male	1 [Reference]		1 [Reference]	
Female	1.315 (1.269–1.362)	<0.001	1.315 (1.269–1.362)	<0.001
Race				
White	1 [Reference]		1 [Reference]	
Asian/Pacific Islander	0.408 (0.384–0.434)	<0.001	0.408 (0.384–0.434)	<0.001
Black	0.670 (0.623–0.721)	<0.001	0.670 (0.623–0.721)	<0.001
American Indian/Alaska Native	2.179 (1.545–3.075)	<0.001	2.179 (1.545–3.075)	<0.001
Pain rating				
I	1 [Reference]		1 [Reference]	
II	0.849 (0.805–0.896)	<0.001	0.849 (0.805–0.896)	<0.001
Total number of in situ/malignant tumors	0.764 (0.740–0.788)	<0.001	0.764 (0.740–0.788)	<0.001
Total number of benign/borderline tumors	1.987 (1.573–2.509)	<0.001	1.987 (1.573–2.509)	<0.001
Cancer site				
Bone and Joint	0.829 (0.416–1.652)	0.59	0.829 (0.416–1.652)	0.59
Breast	1.090 (1.047–1.136)	<0.001	1.090 (1.047–1.136)	<0.001
Digestive	0.784 (0.753–0.816)	<0.001	0.784 (0.753–0.816)	<0.001
Endocrine	1.040 (0.897–1.204)	0.61	1.040 (0.897–1.204)	0.61
Hematological	0.683 (0.597–0.781)	<0.001	0.683 (0.597–0.781)	<0.001
Kaposi Sarcoma	0.435 (0.263–0.722)	0.001	0.435 (0.263–0.722)	0.001
Mesothelioma	0.769 (0.170–3.474)	0.73	0.769 (0.170–3.474)	0.73
Miscellaneous	2.133 (0.484–9.402)	0.32	2.133 (0.484–9.402)	0.32
Respiratory	0.533 (0.486–0.586)	<0.001	0.533 (0.486–0.586)	*p* < 0.001
Skin and Soft Tissue	1.190 (1.114–1.272)	<0.001	1.190 (1.114–1.272)	<0.001
Urogenital	1 [Reference]		1 [Reference]	
Cancer-directed surgery				
Yes	0.902 (0.857–0.951)	<0.001	0.902 (0.857–0.951)	<0.001
No	1 [Reference]		1 [Reference]	
Chemotherapy				
Yes	1.450 (1.370–1.534)	<0.001	1.450 (1.370–1.534)	<0.001
No	1 [Reference]		1 [Reference]	
Survival years after the first tumor diagnosis	1.096 (1.093–1.098)	<0.001	1.096 (1.093–1.098)	<0.001

Abbreviations: OR, odds ratio. ^a^ The ORs are adjusted for age at diagnosis (>20 y), age at death (>65 y), and cancer categories by design and are calculated by univariable logistic regression models. ^b^ The adjusted OR are adjusted for other significant variables apart from the one in every row and calculated by multivariable logistic regression models.

**Table 3 cancers-15-00248-t003:** Sample Characteristics Stratified by Pain Rating ^a^.

Variable	Case Group (*n* = 25,512)			Control Group (*n* = 127,560)		
	I, No. (%) (*n* = 2118)	II, No. (%) (*n* = 23,394)	*p*-Value	I, No. (%) (*n* = 8169)	II, No. (%) (*n* = 11,939)	*p*-Value
Age at cancer diagnosis, mean (SD), y	71.21 (10.11)	71.98 (9.73)	0.001	71.45 (10.06)	71.74 (9.78)	0.01 ^b^
Sex			<0.001			<0.001 ^c^
Male	349 (16.5)	8493 (36.3)		2545 (31.2)	59,104 (49.5)	
Female	1769 (83.5)	14,901 (63.7)		5624 (68.8)	60,287 (50.5)	
Race			0.24			0.72 ^c^
White	1950 (92.1)	21,405 (91.5)		6834 (83.7)	100,123 (83.9)	
Asian/Pacific Islander	86 (4.1)	1131 (4.8)		871 (10.7)	12,383 (10.4)	
Black	80 (3.8)	811 (3.5)		454 (5.6)	6769 (5.7)	
American Indian/Alaska Native	2 (0.1)	47 (0.2)		10 (0.1)	116 (0.1)	
Total number of in situ/malignant tumors	1.22 (0.49)	1.17 (0.45)	<0.001	1.29 (0.59)	1.20 (0.49)	<0.001 ^b^
Total number of benign/borderline tumors	0	0	0.70	0 (0.05)	0 (0.04)	0.26 ^b^
Cancer site			<0.001			<0.001 ^c^
Bone and Joint	0 (0)	10 (0)		0 (0)	65 (0.1)	
Breast	1172 (55.3)	8390 (32.9)		3428 (42)	26,602 (22.3)	
Digestive	372 (17.6)	5037 (19.7)		2636 (32.3)	28,351 (23.7)	
Endocrine	0 (0)	243 (1)		1 (0)	1153 (1)	
Hematological	0 (0)	279 (1.1)		0 (0)	1924 (1.6)	
Kaposi Sarcoma	0 (0)	17 (0.1)		0 (0)	181 (0.2)	
Mesothelioma	0 (0)	2 (0)		0 (0)	15 (0)	
Miscellaneous	0 (0)	3 (0)		0 (0)	7 (0)	
Respiratory	1 (0)	525 (2.1)		12 (0.1)	6858 (5.7)	
Skin and Soft Tissue	390 (18.4)	1428 (5.6)		1317 (16.1)	4240 (3.6)	
Urogenital	183 (8.6)	9578 (37.5)		775 (9.5)	49,995 (41.9)	
Radiation therapy			<0.001			<0.001 ^c^
Yes	311 (14.7)	6196 (26.5)		904 (11.1)	31,437 (26.3)	
No	1807 (85.3)	17,198 (73.5)		7265 (88.9)	87,954 (73.7)	
Cancer-directed surgery			<0.001			<0.001 ^c^
Yes	2115 (99.9)	21,154 (90.4)		8136 (99.6)	104,860 (87.8)	
No	3 (0.1)	2240 (9.6)		33 (0.4)	14,531 (12.2)	
Chemotherapy			<0.001			<0.001 ^c^
Yes	6 (0.3)	1848 (7.9)		28 (0.3)	8585 (7.2)	
No/Unknown	2112 (99.7)	21,546 (92.1)		8141 (99.7)	110,806 (92.8)	
Survival years after the first tumor diagnosis	16.54 (8.81)	15.54 (8.86)	<0.001	12.61 (8.55)	11.65 (8.13)	<0.001 ^b^

Abbreviations: SD, standard deviation. ^a^ Pain rating I indicated in situ behavior code; pain rating II indicated malignant behavior code, with survival years > 1. ^b^
*p*-value by independent t test. ^c^
*p*-value by Χ^2^ test.

## Data Availability

The data presented in this study are available on request from the corresponding author. The data are not publicly available due to privacy.

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
