# Peer review of "Pain as a Protective Factor for Alzheimer Disease in Patients with Cancer"

_cancers, 2022, doi:10.3390/cancers15010248_

Round 1

Reviewer 1 Report

I found the article interesting due to the subject, a very complex one indeed. 

This means that the phases of data collection, variable selection, analysis and interpretation of results are also complex steps. 

My impression is that at the present stage, this work offers a quite limited view of this complexity based on the data that were utilized in support.

This translates into a limited impact, overall, this with reference to any possible conclusion about the nature of the association under study. 

The authors have correctly reported a sub-section of limitations of the work, but unfortunately do not address any of them. 

it would be better to present a detailed statistical analysis and present comparative evaluations among a few methods because this aspect usually affects variable selection, fitting procedure, coding of variables, interactions between variables, relative importance etc. 

it would also be better to substantiate with more evidence the fact that the large study involves strict criteria, risk factors inclusion, subgroup analysis that emphasize the role of digestive cancer pain in AD, reason for some sort of reproducibility to be needed.

I am worried about the lack of information about treatment for AD and pain, possible confounders, but clearly factors associated with variables such as disease stage, type of cancer, cancer site, etc. which therefore generate further unaccounted variability.

Reviewer 2 Report

I really appreciate your work. This study is particularly well done. The topic is relevant. The relationship between cancer and AD is an enigma for which, as the authors point out, has already been the interrogation of an important literature. Your research work pointing the inverse link between cancer pain and AD is an original approach. This is also an unprecedented way  to better understand the pathophysiology of AD

The methodology is correct, although the definition of the 2 pain levels could be better explained.

Presentation of results and analysis is accurate and appropriate. The hypothesis on the association cancer and AD is well developed and offers suitable paths for future research 

Finally, the strengths and limitations of the study are well explained

Round 2

Reviewer 1 Report

The authors have responded to my observations, but the paper remains the same. 

Without additional evidence it is difficult to get to solid conclusions and lot of the reasoning is interesting but not fully validated. 

The paragraph of the limitations and their discussion is defining the current impact of this paper.

I invite the authors to elaborate more on how to concretely address these limitations and link better such detailed aspects with the related concluding remarks.
